# Blind Face Video Restoration with Temporal Consistent Generative Prior and Degradation-Aware Prompt

Submission Id: 908*

## ABSTRACT

Within the domain of blind face restoration (BFR), approaches lacking facial priors frequently result in excessively smoothed visual outputs. Exiting BFR methods predominantly utilize generative facial priors to achieve realistic and authentic details. However, these methods, primarily designed for images, encounter challenges in maintaining temporal consistency when applied to face video restoration. To tackle this issue, we introduce StableBFVR, an innovative Blind Face Video Restoration method based on Stable Diffusion that incorporates temporal information into the generative prior. This is achieved through the introduction of temporal layers in the diffusion process. These temporal layers consider both long-term and short-term information aggregation. Moreover, to improve generalizability, BFR methods employ complex, large-scale degradation during training, but it often sacrifices accuracy. Addressing this, StableBFVR features a novel mixed-degradation-aware prompt module, capable of encoding specific degradation information to dynamically steer the restoration process. Comprehensive experiments demonstrate that our proposed StableBFVR outperforms state-of-the-art methods.

## CCS CONCEPTS

• **Computing methodologies → Computer vision**.

## KEYWORDS

Blind face restoration, diffusion model, facial generative prior, video restoration

**ACM Reference Format:**

Anonymous Author(s). 2024. Blind Face Video Restoration with Temporal Consistent Generative Prior and Degradation-Aware Prompt. In *Proceedings of the 32th ACM International Conference on Multimedia(MM '24), October 28–November 1, 2024,Melbourne, Australia.* ACM, New York, NY, USA, 10 pages. https://doi.org/XXXXXXX.XXXXXXX

## 1 INTRODUCTION

In real-world scenarios, both face images and videos may suffer from unknown and varied types of degradation, such as downsampling, noise, blur, and compression. Blind Face Restoration (BFR) is a challenging task that aims at restoring low-quality faces suffering from unknown degradation. Existing BFR methods usually

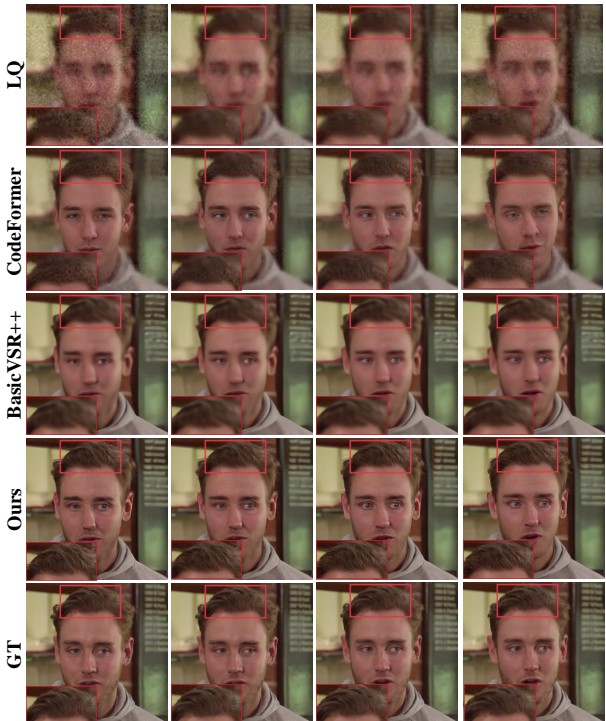

**Figure 1: Four consecutive frames restored by different methods. Blind face image restoration method CodeFormer results in inconsistent visual effects between the face and other regions and temporal inconsistency. Video restoration method BasicVSR++ results in an over-smoothing effect. Our method strikes a good balance between generating texture and temporal consistency.**

use facial priors such as reference prior, geometry prior, and generative prior in the network structure. Among various priors, the generative prior from pre-trained generators, due to its ability to bring more realistic texture and details, has been popularly leveraged by recent BFR methods to restore faces. For example, GFP-GAN [54] incorporates the pre-trained StyleGAN [22] as a decoder into an encoder-decoder architecture. DiffBIR [35] first utilizes a restoration model for preliminary restoration, then introduces Stable Diffusion [45] as generative prior to further refine facial details. Although these existing BFR methods work well in the blind face image restoration (BFIR) problem, they do not fully consider blind face videos. When these BFIR methods are applied to face videos, they usually restore the face cropped from each frame of the video and paste it back into the original frame, and the background is restored using other restoration models such as RealESRGAN [55].

As shown in Fig. 1, this strategy typically leads to two problems: 1) the visual effects of the face and the background are inconsistent; 2) the generated texture is unstable, and the face attributes (*e.g.,* hairstyle, eyes, mouth) may change between frames.

On the other hand, existing video restoration methods achieve temporal consistency by fusing information across frames. For example, BasicVSR++ [7] uses second-order grid propagation and flow-guided deformable alignment to effectively exploit information from the entire input video. VRT [32] adopts transformer architecture for attaining long-range receptive fields. As shown in Fig 1, due to the lack of facial prior, when these methods are used for blind face video restoration, they usually produce over-smooth results that are very inconsistent with human perception. Thus, existing video restoration methods are not applicable for restoring blind face videos. Besides, to the best of our knowledge, there is still no specialized method for restoring blind face videos.

To tackle these challenges, in this paper, we present a Stable Blind Face Video Restoration (StableBFVR). StableBFVR uses the pre-trained latent diffusion model (LDM) Stable Diffusion as facial prior. At the same time, to maintain temporal consistency and use multiple frame information to improve the restoration performance, we introduce temporal layers to Stable Diffusion. These temporal layers comprehensively consider both long-term and short-term information in the video. Specifically, we present Shift-ResBlock which uses the proposed forward temporal shift block and backward temporal shift block alternatively to achieve bi-directional aggregation. The temporal shift blocks first shift input features in the temporal dimension, followed by fusion using convolution blocks. By using Shift-ResBlock repeatedly, the aggregation of long-term information is achieved. For short-term information aggregation, we introduce a Nearby-Frame Attention (NFA). By seeking complementary sharp information existing in neighboring frames, NFA can refine restoration details.

BFR methods usually utilize a wide range of degradation when synthesizing training data. This enhances the generalization ability of the restoration model but also results in a decrease in accuracy. To further improve the restoration performance, we propose a Degradation-Aware Prompt Module (DAPM). DAPM first extracts degradation-aware features from the input frames to predict prompt weights about different types of degradation. Then DAPM utilizes these weights to adjust the corresponding prompt corresponding to different types of degeneration and fuses these prompts to obtain degradation-aware prompts which encode discriminative information about various types of degradation. By interacting with degradation-aware prompts, the StableBFVR can make adaptive responses to various unknown degradations to effectively restore input faces.

Our main contributions can be summarized as follows: (1) We propose StableBFVR which uses generative facial prior to address the blind face video restoration task for the first time. To maintain temporal consistency and improve the restoration performance, we convert pre-trained Stable Diffusion into video restoration models by inserting temporal layers. (2) We present a Degradation-Aware Prompt Module (DAPM) to generate prompts that contain degradation-specific information for dynamically guiding the restoration network. In this way, we can improve the restoration performance and enable the restoration network to adapt to diverse,

unknown degradations. (3) Extensive experimental studies demonstrate StableBFVR achieves SOTA performance on both the public synthetic dataset and real-world low-quality face video dataset we collected from the Internet.

## 2 RELATED WORK

### 2.1 Video Restoration

Video restoration aims to restore high-quality videos from low-quality ones. Most existing video restoration methods can be divided into two categories according to the way they propagate information.

The first [53, 61] usually uses sliding window to aggregate information from adjacent frames to restore the middle single frame. During the alignment stage, they often align all frames in the sliding window towards the middle frame. Earlier methods [5, 62] estimate the optical flow between low-quality neighbouring frames and then perform spatial warping for alignment. Recent approaches employ implicit alignment. For example, some methods [49, 53] align different frames at the feature level with the deformable convolution. Some methods [21, 71] leverage dynamic filters to achieve motion compensation. Some methods [32, 33] mainly use transformer to fuse useful features from adjacent frames. However, multi-frame inputs lead to higher computational complexity and is hard to use larger window sizes to aggregate more distant frames.

The second [6, 7, 33] typically utilizes the recurrent-based method to propagate information from one frame to the next frame, which is accumulated to restore the subsequent frames. These methods usually focuses on designing efficient propagation methods for utilizing longer distance frames. For example, RSDN [19] propose a novel unidirectional propagation with a hidden state adaptation module to enhance robustness to appearance change and error accumulation. Some methods [6, 7] employ bidirectional propagation to better exploit temporal features.

### 2.2 Generative Prior for Blind Face Image Restoration

Early blind face image restoration (BFIR) methods usually employ geometric [4, 8, 9, 65] and reference priors [29–31, 48] to improve the restoration performance. reference priors use the facial component dictionary obtained from additional high-quality face images to guide the face restoration process. Geometric priors use the unique geometric shape and spatial distribution information of faces like facial landmarks, facial heatmaps, and facial parsing maps to help restore high-quality face. However, geometric prior and reference prior are unable to provide rich facial details.

For better visual effects, the generative facial priors from pre-trained generators have been explored for BFIR recently. Some works [54, 63, 73] incorporate the pre-trained StyleGAN [22] as a decoder into an encoder-decoder architecture. Some other works [13, 56, 70] first train VQGAN [11] on high-quality faces with a reconstruction objective, then fine-tune the decoder to adapt to BFIR. Recently, DiffBIR [35] leverages the pre-trained Stable Diffusion (SD) [45] as generative prior which can provide more prior knowledge compared with existing GAN prior and achieves realistic face restoration. Inspired by these works, our approach, for the first time, applies generative priors to the task of blind face video restoration.

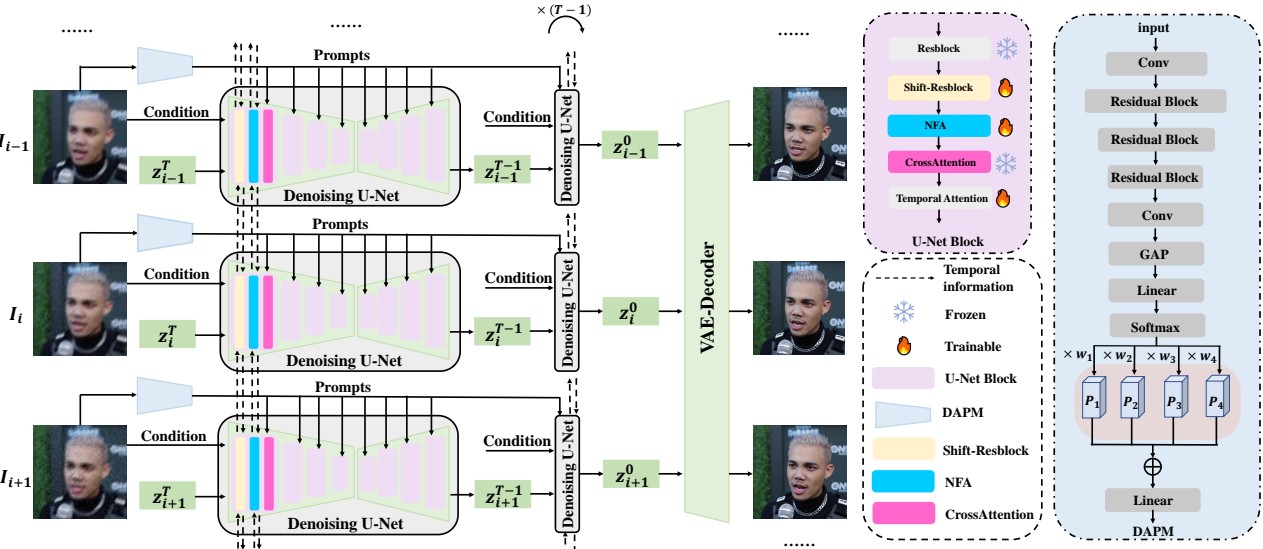

**Figure 2: The architecture of the proposed StableBFVR. We turn Stable Diffusion into a video restoration method by adding temporal layers Shift-Resblock and Nearby-Frame Attention (NFA) into the U-Net block. To further improve performance, we adopt a Degradation-Aware Prompt Module (DAPM) that dynamically guides the diffusion process.**

Moreover, we develop effective techniques to maintain the temporal consistency among continuous frames when restoring facial details.

## 2.3 Diffusion Model

Recently, due to the more stable generation ability than GAN, the diffusion model has been popular in image restoration. Some methods [14, 46, 57] train a diffusion model conditioned on low-quality images and performs restoration through a stochastic denoising process. Some methods [35, 51] fine-tune directly on the pre-trained stable diffusion model to achieve impressive performance. Although the diffusion model has shown promise in image restoration, it is still under-explored in video restoration.

With notable advancements in image generation diffusion model, the number of methods [1, 12, 15, 44] use off-the-shelf image diffusion models with additional temporal layers to achieve video generation. Some methods [16, 17, 47] extend image diffusion models by training them on extensive video pairs. Some methods [24, 36, 52] employ temporal attention mechanisms to generate videos. Some methods [10, 41] propose to introduce optical flow warping in diffusion process. Inspired by video generation works [15, 44, 60] that employ off-the-shelf image diffusion models, our video restoration method exploits pre-trained stable diffusion as a generative prior and proposes a novel temporal strategy, resulting in temporal consistency.

## 2.4 Prompt Learning

With the extensive application of prompt learning in the field of NLP [3, 37] and high-level vision tasks [18, 20], prompt learning has recently also been widely used in image restoration to better utilize the degradation context, such as the all-in-one restoration

tasks [38, 43]. Although prompt learning performs well in all-in-one restoration tasks, the degraded image contains only a single type of degradation. Our approach for the first time explores the application of prompt learning in dealing with mixed degradation tasks like the case of blind face restoration.

## 3 METHODOLOGY

BFR methods can use pre-trained generation models to restore high-quality images with clear facial details. However, if we directly use generative prior for video restoration, the inherent stochastic nature of the generation model leads to temporal inconsistencies in the restored video. Especially for face videos, in addition to the flickering artifacts, it also causes the face attributes (*e.g.,* hairstyle, eyes, mouth) in the restored video to be inconsistent.

By training on a massive amount of high-quality images, Stable Diffusion has powerful prior knowledge about face images. Our objective is to harness the knowledge from Stable Diffusion for blind face video restoration. As shown in Fig. 2, we introduce temporal layers in the Stable Diffusion to preserve temporal consistency. First, we propose Shift-Resblock which implicitly captures global information for long-term aggregation. Second, we further improve restoration performance and temporal consistency by introducing Nearby-Frame Attention to aggregate short-term information. Moreover, to enable adaptive responses to complex and large-range blind degradation, we propose a degradation-aware prompt module to encode degradation-specific information as prompts to guide the restoration network.

### 3.1 Preliminary: Latent Diffusion Models

**Pre-trained Stable Diffusion** Stable Diffusion employs the LDM framework. It utilizes the encoder of Variational Autoencoders (VAE) to map the image $x$ into the latent $z$ to perform the diffusion

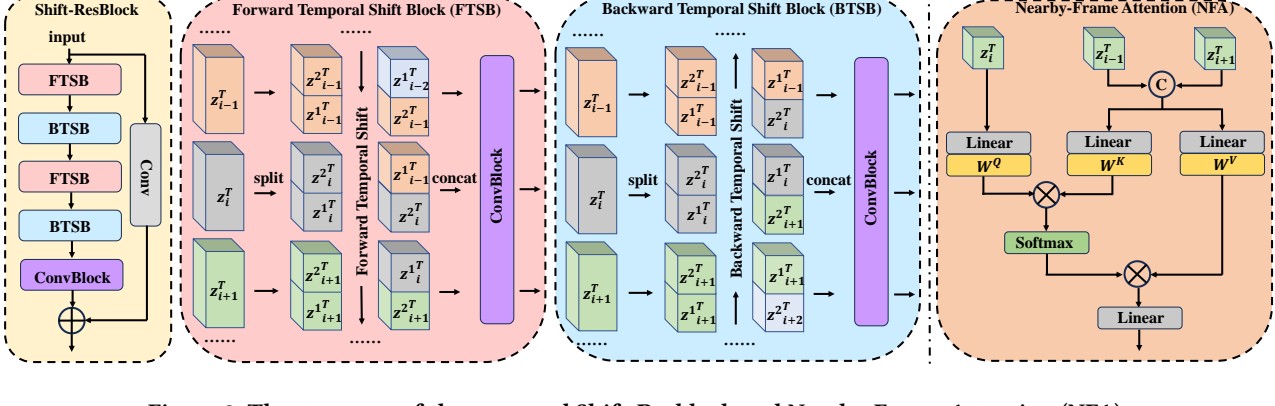

**Figure 3: The structure of the proposed Shift-Resblock and Nearby-Frame Attention (NFA).**

and denoising processes, then reconstruct it with the decoder of VAE. In the diffusion process, the diffused latent $z_t$ can be directly generated by applying Gaussian noise with variance $\beta_t \in (0, 1)$ to the latent $z$ at any time step $t$. This process can be described as:

$$z_t = \sqrt{\bar{\alpha}_t}z + \sqrt{1 - \bar{\alpha}_t}\epsilon, \tag{1}$$

where $\alpha_t = 1 - \beta_t$, $\bar{\alpha}_t = \prod_{i=1}^{t} \alpha_i$ and $\epsilon \sim \mathcal{N}(0, I)$ is a random Gaussian noise. In the reverse process, the U-Net denoiser $\epsilon_\theta$ parameterized by $\theta$ is trained to predict the noise $\epsilon$. The optimization objective of the latent diffusion model can be defined as follows:

$$\mathcal{L} = \mathbb{E}_{z_t,t,c,\epsilon}[||\epsilon - \epsilon_\theta(z_t, t, c)||_2^2], \tag{2}$$

where $t$ is a randomly selected time-step, $c$ is an optional condition (e.g., text, images, and representations), and $\epsilon$ is sampled from the standard Gaussian distribution.

In this work, we start with the pre-trained Stable Diffusion and create a new video diffusion model for blind face video restoration. By adopting temporal strategies within the LDM framework, our method can achieve temporal consistency while leveraging the prior knowledge from Stable Diffusion.

### 3.2 Temporal Layers in StableBFVR

To apply the pre-trained Stable Diffusion to video-related tasks, we propagate both long-term and short-term temporal information among different input frames to maintain temporal consistency. Long-term information helps preserve face attribute consistency among long-range frames. Short-term information relieves flickering artifacts of adjacent frames. Considering that the degree of degradation between different frames of the video is different, propagating temporal information also helps improve the restoration performance.

**Long-term Information Aggregation.** Previous works [10, 41] demonstrate the benefits of optical flow-guided long-term propagation in video diffusion models. Considering that blind degraded video usually contains severe blur or noise, this kind of degradation will affect the accuracy of the optical flow estimation network [28, 66, 72], subsequently leading to poor performance of the optical flow-guided restoration network. This suggests that optical flow is not suitable for our task. Prior works [34, 40] have demonstrated that temporal shift operation can blend information from other frames with the current frame along the temporal dimension

and establish the temporal correspondences implicitly. Thus, we introduce the Shift-Resblock in the basic U-Net blocks to effectively establish temporal correspondences and conduct long-term fusion.

As shown in Fig. 3, supposing the $i$-th frame input feature of Shift-Resblock at time step $T$ is $Z_i^T \in \mathbb{R}^{C \times H \times W}$, Shift-Resblock consists of Forward Temporal Shift Block (FTSB) which fuses the feature of $(Z_{i-1}, Z_i)$ and a Backward Temporal Shift Block (BTSB) which fuses the feature of $(Z_i, Z_{i+1})$. By stacking FTSB and BTSB alternatively, Shift-Resblock can achieve bi-directional aggregation. Although a single Shift-Resblock can only fuse adjacent frame information, we can achieve long-term aggregation by using Shift-Resblock in our framework repeatedly. In the temporal shift, each $Z_i^T$ is split into two parts $Z1_i^T \in \mathbb{R}^{C^1 \times H \times W}$ and $Z2_i^T \in \mathbb{R}^{C^2 \times H \times W}$ along the channel dimension. In the forward temporal shift, we shift the feature $Z1_{i-1}^T$ from the $(i-1)$-th frame to the $i$-th frame, then feature $Z1_{i-1}^T$ and feature $Z2_i^T$ are merged as feature $Zf_i^T$ of the $i$-th frame. The output of the forward temporal shift for the $i$-th frame can be defined as:

$$Zf_i^T = Concat(Z1_{i-1}^T, Z2_i^T), 0 < i \le F, \tag{3}$$

where $F$ is the number of input frames. In particular, in the forward temporal shift, the first frame remains unchanged. In the backward temporal shift, we shift the feature $Z2_{i+1}^T$ from the $(i+1)$-th frame to the $i$-th frame, then feature $Z2_{i+1}^T$ and feature $Z1_i^T$ are merged as feature $Zb_i^T$ of the $i$-th frame. The output of the backward temporal shift for the $i$-th frame can be defined as

$$Zb_i^T = Concat(Z1_i^T, Z2_{i+1}^T), 0 \le i < F. \tag{4}$$

Analogously, in the backward temporal shift, the last frame remains unchanged. After the temporal shift, we utilize a simple convolution block independently on each frame to capture and aggregate both the spatial and temporal information.

Moreover, following the existing methods [1, 58], we also introduce temporal attention to the U-Net blocks. The temporal attention performs self-attention along the temporal dimension for temporal modeling.

**Short-term Information Aggregation.** The original U-Net block has a spatial self-attention. When the input is multiple frames, it acts only on each frame alone. Considering that the short-term

**Table 1: Quantitative comparison on VFHQ-Test and WebVideo-Test for blind face video restoration. Red and Blue indicate the best and the second-best performance.**

| | VFHQ-Test | | | | | | | WebVideo-Test | | |
|---|---|---|---|---|---|---|---|---|---|---|
| | LPIPS↓ | NIQE↓ | MUSIQ↑ | CLIP-IQA↑ | WE↓ | PSNR↑ | SSIM↑ | NIQE↓ | MUSIQ↑ | CLIP-IQA↑ |
| Input | 0.4591 | 10.237 | 16.35 | 0.276 | 13.66 | 25.84 | 0.7564 | 8.361 | 37.08 | 0.286 |
| GFPGAN [54] | 0.4139 | 5.587 | 65.86 | 0.601 | 16.10 | 26.30 | 0.7624 | 4.897 | 72.01 | 0.615 |
| RestoreFormer [56] | 0.4162 | 5.615 | 59.86 | 0.583 | 16.73 | 26.17 | 0.7504 | 4.842 | 68.08 | 0.628 |
| CodeFormer [70] | 0.4116 | 5.603 | 64.04 | 0.604 | 16.24 | 26.32 | 0.7588 | 4.991 | 70.55 | 0.640 |
| DiffBIR [35] | 0.4354 | 7.293 | 53.15 | 0.512 | 15.88 | 26.38 | 0.7603 | 6.012 | 63.78 | 0.581 |
| BaiscVSR++ [7] | 0.3406 | 9.149 | 50.15 | 0.294 | 6.28 | 28.05 | 0.8213 | 7.803 | 54.88 | 0.299 |
| DSTNet [42] | 0.3493 | 9.641 | 43.66 | 0.297 | 6.27 | 28.30 | 0.8319 | 8.340 | 53.66 | 0.350 |
| RVRT [33] | 0.3710 | 9.419 | 37.81 | 0.246 | 6.21 | 27.79 | 0.8104 | 8.316 | 45.57 | 0.272 |
| **Ours** | 0.3119 | 5.262 | 75.33 | 0.759 | 13.45 | 26.58 | 0.7689 | 4.512 | 74.20 | 0.693 |
| GT | 0 | 4.778 | 72.83 | 0.645 | 7.10 | ∞ | 1 | - | - | - |

adjacent frames are usually highly similar to the current frame, they can provide sufficient information for the restoration of the current frame. Thus, to further enhance the restoration performance, we present a Nearby Frame Attention (NFA) mechanism that extends the spatial self-attention to the temporal domain. By seeking complementary sharp information in neighboring frames, NFA can capture spatio-temporal consistency. The structure of NFA is shown in Fig. 3. Specifically, given the $i$-th frame feature maps $Z_i \in \mathbb{R}^{C \times H \times W}$ as input, our NFA takes current frame input features $Z_i$ as query $Q = W_q(Z_i)$, while the key $K = W_k(Concat(Z_{i-1}, Z_{i+1}))$ and value $V = W_v(Concat(Z_{i-1}, Z_{i+1}))$ are generated from the concatenation of the former frame and the latter frame, where $W_q, W_k, W_v$ are projection matrices shared across space and time. Finally, we adopt the self-attention mechanism to conduct short-term information aggregation. The output is a weighted sum of the value, weighted by the similarity between the query and key features. Note that, in the training process, we only update the parameters of the query process, and the other parts of the parameters are frozen.

### 3.3 Degradation-Aware Prompt Module

Previous work [68] has proven that when using complex large-scale degradation to train blind face restoration methods, it will enhance the generalization ability, but at the cost of decreasing accuracy. To address this problem, we propose a degradation-aware prompt module (DAPM) to generate prompts that can dynamically adjust the prediction of the degree of degradation of the input frames. It can help the restoration network make adaptive responses to various unknown degradations.

The structure of DAPM is shown in Fig 2. Considering that blind degradation usually consists of four different degradations: blur, noise, compress, and downsample, we establish a degradation set $P = \{P_1, P_2, P_3, P_4 \mid P_N \in \mathbb{R}^{L \times C}\}$ to encapsulate information for different degradation. Serving as learnable parameters, $P$ can interact with the dynamical weights that are predicted from the input degraded frame. Thus it can function as prompts aware of degradations.

Specifically, to predict dynamical weights, DAPM first extracts features $F_i^0 \in \mathbb{R}^{C \times H \times W}$ from a given degraded input frame $I_i$ by applying a convolution operation. Then the feature is sent to a

three-level encoder, with each level of the encoder employs several residual blocks. The feature will be transformed into the compact feature $F_i^1 \in \mathbb{R}^{4C \times \frac{H}{4} \times \frac{W}{4}}$ which is rich in degradation-aware information. Then we apply global average pooling (GAP) across the spatial dimension to generate a feature vector $V_i \in \mathbb{R}^C$. Next, we use a linear layer and softmax operation to obtain prompt weights $w_1, w_2, w_3, w_4$ about the four kinds of degradation. Finally, considering that different degradation is not independent, degradation will also affect each other. After we use these weights to make adjustments in the degradation set $P$, we use a linear layer to fuse them. The process of generating degradation-aware prompts is

$$w_i = Softmax(Linear(GAP(F_i^1))), \tag{5}$$

$$\hat{P} = Linear(\sum_{i=1}^{4} w_i P_i). \tag{6}$$

The generated degradation-aware prompt will be fed into CrossAttention in the denoising U-Net block to dynamically guide the restoration network.

## 4 EXPERIMENT

### 4.1 Datasets and Implementation

**Training Datasets.** We train our method on $2,000$ clips randomly chosen from VFHQ [59]. VFHQ is a high-quality video face dataset, which contains over $16,000$ high-fidelity clips of human faces. We resize all the frames to $512 \times 512$ during training. We train our method on synthetic data that approximate to the real low-quality images. Similar to the common practice in blind face restoration [54, 70], the degradation model is as follows:

$$y = [(x \circledast k_\sigma) \downarrow_r + n_\delta]_{FFMPEG_{crf}}, \tag{7}$$

where $y$ is the synthetic low-quality frame, $x$ is the high-quality frame, $k_\sigma$ is Gaussian blur kernel, $r$ represents the down-sample factor, and $n_\delta$ is white Gaussian noise. We incorporate the $FFMPEG_{crf}$ compress into the degradation model, where $crf$ is the constant rate factor that decides how many bits will be used for each frame. Compared with the $JPEG$ compress used in BFIR, $FFMPEG_{crf}$ can implicitly consider the inter-dependencies between frames, providing temporal and spatial degradations. For each training pair,

**Figure 4: Visual comparison results of different methods on the VFHQ-Test. Our StableBFVR produces more faithful details. Zoom in for best view.**

we randomly sample $\sigma$, $r$, $\delta$, and $crf$ from $[0.1, 10]$, $[1, 4]$, $[0, 15]$, $[18, 25]$, respectively.

**Testing Datasets.** We evaluate our method on synthetic dataset **VFHQ-Test** and real-world dataset **WebVideo-Test**. They both have no overlap with our training dataset. **VFHQ-Test** is composed of 50 high-quality clips. We choose the first 100 frames of each clip, a total of $5,000$ frames as our test set. To synthesize testing pairs, we apply the same degradation model as the training phase. To better evaluate the generalization of blind face video restoration methods in the real world, we propose a real-world test set **WebVideo-Test**. The videos in our WebVideo-Test dataset are collected from video websites. It consists of 10 video clips, each containing 100 frames of diverse and complicated degradation.

**Evaluation Metrics.** For evaluation of the VFHQ-Test with ground truth, we adopt pixel-wise metrics PSNR and SSIM and perceptual metric LPIPS [69]. We also employ widely-used non-reference perceptual metrics NIQE [39], CLIP-IQA [50], and MUSIQ [23]. For the real-world dataset WebVideo-Test without ground truth, we adopt only the three non-reference metrics mentioned above. Compared with BFIR, one major aspect of the BFVR problem is the temporal consistency of the restored videos. In this work, we adopt the average warping error (WE) [26] of the restored videos to quantitatively measure the temporal consistency. It can be calculated as:

$$WE = \frac{1}{N-1} \sum_{i=2}^{N} ||\hat{I}_i - \mathbf{W}(I_{i-1}, S_{i-1->i})||_1, \tag{8}$$

where $\hat{I}_i$ is the predicted frame, $\mathbf{W}$ denotes the spatial warping operation, and $S_{i-1->i}$ is the estimated optical flow from ground-truth video. We use $10^{-3}$ quantity level when showing this metric.

**Implementation Details.** We utilize Stable Diffusion V2.1 to initialize the weight of our StableBFVR. Then we fix the weight of our StableBFVR except for the proposed components and the condition. Regarding the condition, we first employ frozen pre-trained BasicVSR++ for preliminary restoration, then adopt trainable ControlNet [67], initialized with the weight from BFIR method DiffBIR [35], encode the input frame as a condition and inject it into the denoising U-Net. The training is conducted on 4 NVIDIA A100 GPUs, with batch size 4 and the number of input frames 8. The learning rate is set to $1 \times 10^{-4}$ using the Adam [25] optimizer and we train it for $100K$ iterations. During inference, we divide the low-quality video into multiple sequences. For each sequence, the number of input frames is set to 32 and we run the sampling for 50 steps.

## 4.2 Results

We compare our StableBFVR with several state-of-the-art methods, including four BFIR models, GFPGAN [54], Restoreformer [56], CodeFormer [70], DiffBIR [35], and three video restoration models, BasicVSR++ [7], DSTNet [42], RVRT [33]. For BFIR models, we adopt their officially released models in the experiments. Following original implementations in video restoration, the BFIR model only restores the face detected in the video frame, while the background is enhanced by RealESRGAN [55]. For video restoration models, to

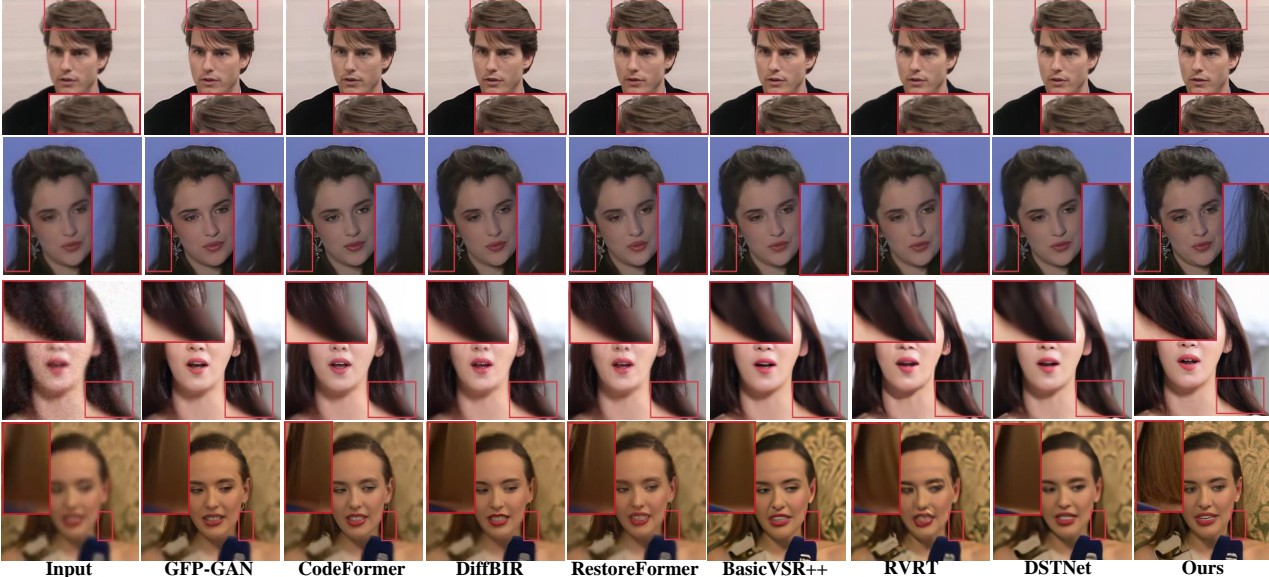

**Figure 5: Visual comparison results of different methods on the real-world dataset WebVideo-Test. Our StableBFVR produces more faithful details. Zoom in for best view.**

ensure a fair comparison, these methods are re-trained under the same training settings.

**Quantitative Comparison.** For the synthetic VFHQ-Test, the quantitative results are shown in Tab. 1. The results indicate that our method achieves state-of-the-art performance on all perceptual metrics. Specifically, our StableBFVR achieves the best performance regarding LPIPS, indicating that the perceptual quality of restored face videos is closest to ground truth. Moreover, StableBFVR also obtains the best results of NIQE, MUSIQ, and CLIP-IQA, showing that the outputs better align with human visual and perceptive systems. Note that, like other BFIR methods that use generative prior, our model is also not strong at PSNR and SSIM. Because PSNR and SSIM do not correlate well with the human visual and perceptive systems [2, 27]. In general, over-smoothing images will derive higher PSNR and SSIM values. The methods based on generative priors produce more high-frequency texture details, resulting in lower PSNR and SSIM.

To assess the generalization ability, we extend the evaluation of our model to the real-world dataset WebVideo-Test. The quantitative results are presented in Tab. 1. StableBFVR exhibits superior performance across all three metrics NIQE, MUSIQ, and CLIP-IQA, showing its remarkable generalization capability. Furthermore, compared with video restoration methods, BFIR methods also show satisfactory performance, suggesting the importance of generative prior in the scenery of unknown degradations in the real world.

**Qualitative Comparison.** For the synthetic VFHQ-Test, the qualitative results are illustrated in Fig. 4. Compared with video restoration methods, thanks to the powerful generative facial prior, our method recovers faithful details in the eyes, mouth, beard *etc*. On the contrary, face videos restored by video restoration methods are over-smooth and lose facial texture details. BFIR methods only restore the face detected in the video frame, with the background

restored by RealESRGAN. Consequently, face videos restored by BFIR exhibit inconsistent visual effects between the face region and background region, even showing obvious boundaries (3rd row, Fig. 4). In contrast, our method treats the input as a whole in restoration and performs well in all regions. In addition, compared with BFIR methods, our method can aggregate information from other frames to improve performance. As a result, in scenarios where BFIR methods exhibit poor performance, our method can still generate superior facial details (4th row, Fig. 4).

For WebVideo-Test, the qualitative results are shown in Fig. 5. Our method produces realistic facial textures in the case of complicated real-world degradation. As shown in the last column of Fig. 5, previous BRIR methods fail to restore the hair textures on the image boundary, while ours is successful. Compared with video restoration methods, our method produces significantly more texture detail.

**Temporal Consistency.** The quantitative assessment of temporal consistency is presented in Tab. 1. It is worth mentioning that the metric WE may not be able to faithfully reflect the human perception of the temporal consistency [64]. For example, over-smoothing sequences usually have much higher WE scores despite unpleasant perceptual quality. As shown in Tab. 1, the scores of the video restoration methods are even higher than the ground truth. Given that our StableBFVR tends to generate more details and textures, which adversely impact the WE value, it exhibits a less favorable performance in this regard. Nonetheless, StableBFVR still outperforms other BFIR methods driven by the generative facial priors.

To thoroughly verify our method, we visualize the consecutive frames generated by different methods in Fig. 6. It is observed that, although sequences restored by BFIR methods exhibit realistic texture, there are noticeable differences between the textures of

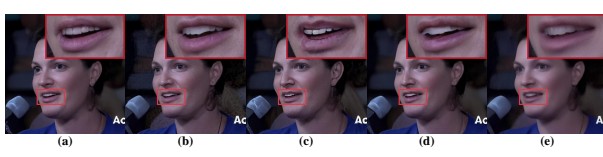

**Figure 6: Visual comparisons of the temporal consistency for restored videos. (a) GFP-GAN, (b) CodeFormer, (c) DiffBIR, (d) RestoreFormer, (e) BaiscVSR++, (f) RVRT, (g) DSTNet, (h) Ours. Zoom in for best view.**

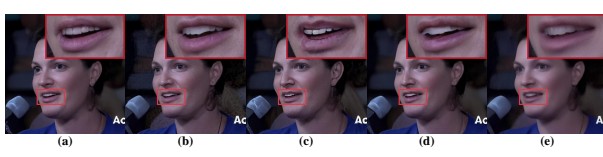

**Figure 7: Visual comparison results of ablation study. (a) Full Method, (b) w/o Shift-Resblock, (c) replace NFA with SA, (d) replace DAPM with CLIP, (e) w/o DAPM. Zoom in for best view.**

**Table 2: Ablation studies of our StableBFVR on VFHQ-Test.**

| Configuration | LPIPS↓ | MUSIQ↑ | WE↓ |
|---|---|---|---|
| replace NFA with SA | 0.3312 | 73.18 | 14.89 |
| w/o Shift-Resblock | 0.3541 | 72.33 | 15.44 |
| w/o DAPM | 0.3263 | 74.69 | 13.48 |
| replace DAPM with CLIP | 0.3289 | 74.86 | 13.52 |
| inference frames 8 | 0.3162 | 74.82 | 14.91 |
| inference frames 16 | 0.3144 | 75.19 | 13.96 |
| inference frames 24 | 0.3120 | 75.23 | 13.65 |
| Full method | 0.3119 | 75.33 | 13.45 |

continuous frames. Conversely, sequences restored by video restoration methods demonstrate commendable temporal consistency but tend to be excessively smooth, lacking textures. StableBFVR strikes a favorable balance, reconstructing more textures, while simultaneously preserving temporal consistency.

### 4.3 Ablation Studies

**Effectiveness of Temporal Layers.** As depicted in Tab. 2, we explore the significance of the temporal layers. Specifically, we first remove Shift-ResBlock, resulting in a noticeable decline in both temporal consistency and perceptual metrics. The impact of this configuration change is shown in Fig. 7, where not only are the details of the teeth inadequately restored, but artifacts also emerge in the background. It implies that Shift-Resblock can improve perceptual quality and consistency through the aggregation of long-term information. Subsequently, we replace NFA with the original

Self-Attention from Stable Diffusion. This replacement leads to a performance decrease in terms of both temporal consistency and perceptual metrics. Fig. 7 illustrates that, while the texture of the teeth can be generated, the quality is notably diminished. This observation emphasizes that the rich contextual information from neighboring frames extracted by NFA plays a crucial role in refining details.

**Effectiveness of DAPM.** We then investigate the significance of the DAPM. We first directly remove DAPM. This alteration leads to a drop in perceptual performance. Subsequently, we replace DAPM with CLIP. Following other Stable Diffusion-based restoration methods [35, 51], we set the input of CLIP as an empty string. This substitution similarly leads to a reduction in perceptual performance. Fig. 7 visually demonstrates that both configurations fail to restore the texture of teeth, implying the instrumental role of DAPM in enhancing restoration.

**The Number of Inference Frames.** As detailed in Tab. 2, we observe that the input number of frames during inference can affect the performance. The more the input frames are, the better the performance is. Especially for temporal consistency, there is a significant enhancement when the number of input frames increases from 8 to 16. These results also illustrate that propagating information about distant frames helps improve restoration performance and temporal consisten.

## 5 CONCLUSION

In this work, we tackle the BFVR problem for the first time. We propose StableBFVR leveraging the strong generative prior from the pre-trained generative model Stable Diffusion to restore face videos with realistic details. To ensure content consistency among frames and use multi-frame information for improved restoration, we develop Shift-Resblock and Nearby-Frame Attention to aggregate both long-term and short-term information. Additionally, we propose a Degradation-Aware Prompt Module to dynamically guide the restoration process and further enhance performance. Extensive experiments show that our StableBFVR achieves superior performance than video restoration methods and blind face image restoration methods.

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
