# OpenReview forum: "Blind Face Video Restoration with Temporal Consistent Generative Prior and Degradation-Aware Prompt"
_acmmm.org/ACMMM/2024/Conference — MM2024 Poster_

### Official Review · Reviewer_qzux · 2024-05-22

**Rating:** 4
**Confidence:** 3

**Summary:**

This is a work on Blind Face Video Restoration. In the past, the most common practice in video restoration tasks was to split down the aligned face, repair the face and the rest of the background separately, and then put together the restoration results. The article suggests that this way of restoring aligned faces by segmenting them separately in the past tends to cause inconsistency in the texture of the face region and the rest of the background, which is more obvious in video sequences.

**Strengths:**

(1) A combination of longterm information and short-term information is used to maintain the inter-frame consistency of the video. The structure in Fig. 3 is very clear and specific.

(2) The experimental results are very sufficient and the various comparisons are very clear.

**Limitations:**

(1) NFA looks like Cross-Attention applied inside the front and back frames to improve the temporal consistency, which is also reflected by the WE metrics in the ablation experiments in Table 2, is there any comparison of the video sequences on the visualization results in Figure 7? To illustrate the improvement of this module on the consistency of front and back information.

(2) Nearby Frame Attention (NFA) is very similar to cross-attention, may I ask what is the difference and benefit here?

(3) In Degradation-Aware Prompt Module (DAPM), it is mentioned that “DAPM first extracts degradation-aware features from the input frames to predict prompt weights about different types of degradation”. May I ask how to judge whether the weights and types predicted here are accurate or not, since the types need to be set in advance? Can the weights here be reversed to do degradation on result images to see if the prediction is accurate? And will this way of setting generalize the real-world data poorly?

(4) The article doesn't answer the question of not-aligned (line 118), if we just use the not-aligned FACE dataset and SD's a priori to sort of solve the doubts, I feel that it's not quite complete, and there's no direct comparison of the reasonableness of the exposition.

(5) The WE calculation for the experimental metrics uses the optical flow of GT, is this not quite right for the continuity metric of the RESULT video?

**Suitability:**

3

---

### Official Review · Reviewer_spet · 2024-05-25

**Rating:** 4
**Confidence:** 3

**Summary:**

This paper uses Stable Diffusion as prior for BFVR(Blind Face Video Restoration) task, introduces StableBFVR pipeline with additional temporal blocks for temporal consistency.

**Strengths:**

1. The paper introduces SD to BFVR task, achives better details such as hair or background area.
2. All BFVR methods compared are retrained use the same degradation settings.

**Limitations:**

1. The time consistency(WE) is worse than other VSR methods, which may increase the sense of strangeness when watching the restored video.
2. From the ablation study, DAPM plays small role in the process.

**Suitability:**

3

---

### Official Review · Reviewer_zZ68 · 2024-05-25

**Rating:** 4
**Confidence:** 3

**Summary:**

This paper presents a novel method called StableBFVR for Blind Face Video Restoration using a Stable Diffusion-based framework. It innovatively integrates generative prior, temporal consistency and Degradation-Aware Prompt Module (DAPM) to handle video restoration from unknown types of degradations. The method utilizes temporal layers within the diffusion process, specifically designed Shift-ResBlock and Nearby-Frame Attention (NFA) to manage long-term and short-term information effectively. In experiments part, it demonstrates that StableBFVR outperforms existing state-of-the-art methods in terms of realism and detail in restored videos.

**Strengths:**

On the methodological level, the incorporation of Shift-ResBlock and Nearby-Frame Attention (NFA) mechanisms substantially enhances temporal consistency in video restoration tasks. Concurrently, the Degradation-Aware Prompt Module (DAPM) significantly improves the model's ability to generalize across a diverse array of real-world degradations. Moreover, the employment of a diffusion-based generative prior gives richer details, resulting in output quality outperforming the ground truth in some evaluation metrics. On the implementation level, multiple evaluation metrics and comparative methods are set up, and tests are conducted on web videos. It demonstrates the stability of StableBFVR.

**Limitations:**

1.	The specific contribution of DAPM has not been explicitly detailed. Given that DAPM is capable of detecting the proportional contribution of four components, it would be prudent to design experiments to illustrate its effectiveness, when degradations are artificial.
2.	While the paper demonstrates superior performance on various benchmarks, it lacks a detailed discussion on potential failure cases or limitations. Particularly when employing a diffusion prior, the images generated may exhibit certain deviations from the ground truth. Are the details necessarily superior?
3.	There are certain issues at the level of paper composition. From a qualitative perspective, the images in the comparison module are too small, making it difficult to effectively showcase the content optimized by StableBFVR in some cases. Additionally, there are some typographical errors in the paper, such as the capitalization issue in line 214.

**Suitability:**

2

---

### Official Review · Reviewer_WjiR · 2024-05-26

**Rating:** 5
**Confidence:** 2

**Summary:**

The topic of this paper is blind face video restoration. This paper introduces temporal layers into diffusion to aggregate long-term and short-term information, and introduce mixed-degradation aware prompt module to encode degradation information.

**Strengths:**

1.  This paper aims to address temporal consistency and generalizability issues in blind face video restoration.
2.  This paper presents corresponding strategies.

**Limitations:**

1.  This paper claims that existing BFR methods employ complex methods. I suggest that the authors analyze the complexity of the proposed model, as it involves information aggregation and mixed-degradation prompt module. I think these modules also increase the complexity.
2.  This paper evaluates the proposed method on two datasets, it seems limited for comprehensive evaluation.
3.  Table 1 does not provide error bars. I suggest that the authors give std.
4.  I think the authors should analyze why these prompts can aware of degradation.
5. The code should be released.

**Suitability:**

3

---

### Meta-Review · Area_Chair_dBYZ · 2024-07-02

**Recommendation:** Accept (Poster)
**Confidence:** 5

**Metareview:**

The paper presents a well-executed approach with strong experimental validation and practical relevance. However, the incremental nature of some innovations, increased complexity, and limited detailed explanations warrant caution. Addressing these issues through minor revisions could significantly improve the paper's quality and impact.